# Selection Behavior and OBP-Transcription Response of Western Flower Thrips, *Frankliniella occidentalis*, to Six Plant VOCs from Kidney Beans

**DOI:** 10.3390/ijms241612789

**Published:** 2023-08-14

**Authors:** Yanhui Wang, Xiaobing Zhu, Yixuan Jin, Ruichuan Duan, Yunkai Gu, Xiaowei Liu, Lei Qian, Fajun Chen

**Affiliations:** 1Department of Entomology, College of Plant Protection, Nanjing Agricultural University, Nanjing 210095, China; 2Institute of Leisure Agriculture, Jiangsu Academy of Agricultural Sciences, Nanjing 210014, China

**Keywords:** *Phaseolus vulgaris* L., *Frankliniella occidentalis*, VOCs, attractive and repellant effects, selection behavior

## Abstract

Plant volatile organic compounds (VOCs) are an important link that mediates chemical communication between plants and plants, plants and insects, and plants and natural enemies of insect pests. In this study, we tested the response in the selective behavior of western flower thrips, *Frankliniella occidentalis*, to the VOCs of kidney bean, *Phaseolus vulgaris* L., to explore their “attraction” or “repellent” effects regarding their application in integrated pest management (i.e., IPM). The results indicated that 12.7 μL/mL (E, E, E, E)-squalene, 3.2 μL/mL dioctyl phthalate, and 82.2 μL/mL ethyl benzene had a significantly attractive effect on the selective behavior of *F. occidentalis*, while 10.7 μL/mL and 21.4 μL/mL 2,6-ditert-butyl-4-methyl phenol had a significantly repulsive effect on the selective behavior of *F. occidentalis*, showing that *F. occidentalis* responds differently to specific concentrations of VOCs from *P. vulgaris* plant emissions. Interestingly, the three compounds with the specific above concentrations, after being mixed in pairs, significantly attracted *F. occidentalis* compared to the control treatment; however, the mixture with the three above compounds had no significant different effect on *F. occidentalis* compared to the control treatment. It can be seen that the effect with the mixtures of three kinds of VOCs had the same function and may not get better. Simultaneously, the reasons for this result from the transcription levels of odorant-binding protein genes (OBPs) were determined. There were differences in the types and transcription levels of OBPs, which played a major role in the host selection behavior of *F. occidentalis* under the mixed treatment of different VOCs. It is presumed that there are specific VOCs from *P*. *vulgaris* plants that have a good repellent or attracting effect on the selective behavior of *F*. *occidentalis*, which can be used for the development of plant-derived insect attractants and repellents to serve as IPM in fields. But attention should be paid to the antagonism between plant-derived preparations and VOCs produced by plants themselves after application.

## 1. Introduction

Western flower thrips, *Frankliniella occidentalis* (Pergande) (Thysanoptera: Thripidae) is an important worldwide invasive herbivorous insect pest in agriculture, which has successfully invaded many countries since the 1970s [1,2]. *F. occidentalis* causes damage by direct feeding and oviposition on leaves, flowers, and fruits, or by indirectly transmitting plant viruses [3]. And it feeds on over 240 plant species, including numerous ornamental crops and vegetables, as well as on other plants [4], and causes substantial economic losses in many greenhouse and field crops [3]. In addition, thrips pierce host plant cells and suck out the contents from narrow crevices within or between plant parts by using rasping-sucking mouthparts, which differs from the well-studied caterpillars or aphids [5]. At present the main control method of *F. occidentalis* is using chemical pesticides such as imidacloprid, ethyl spinosad, poisoning spleen, emamectin benzoate, methomyl, and avermectin [6,7,8], but the “3R” effects (i.e., Resistance, Resurgence, and Residue) produced by the abuse of chemical pesticides of *F. occidentalis* have become more and more obvious for a long time [4]. Many studies have reported that *F. occidentalis* has developed different degrees of resistance to organophosphorus, organochlorine, and pyrethroid insecticides [9,10,11]. In addition, it is more and more difficult to use chemical reagents to effectively control *F. occidentalis* as they can hide easily and have the advantage of tiny body size [12,13]. The development of new biological agents or the integration of multiple biological controls is designed to reduce insecticide inputs, thus diminishing selection pressure placed on *F. occidentalis* populations, and decreasing insecticide resistance [7].

Complex interactions form between host plants, herbivorous insect pests, and natural enemies in a long-term coevolution process, and plant volatile organic compounds (VOCs) are an important way for information exchange between/among them. Herbivorous insects can locate host plants, and natural enemy insects can locate host insects, by VOCs, and plants can also release volatile substances to repel herbivorous insect pests as a direct defense. The long-term interaction and co-evolution with herbivorous insects and host plants lead to the attraction and repelling effects of VOCs on insects, interfering with their ability to develop resistance to plant volatiles [14,15]. The olfactory organs of insects can identify plant volatile secondary metabolites to determine suitable host plants. For example, the volatiles γ-nonolactone, γ-octanolactone, and 6-pentyl-2H-pyran-2-one are released after nectarine ripening attracted *Thrips obscuratus* [16]. The volatiles of tobacco plants damaged by *Heliothis virescens* can repel *F. occidentalis*, which might be due to the defensive volatiles released by tobacco plants after being attacked, or it might be that *F. occidentalis* recognized the existence of competitors through the tobacco volatiles [17]. The single components of rice volatiles, α-pinene, methyl salicylate, linalool, basil, trans-2-hexenal, and leaf alcohol, attracted the natural enemy of *Nilaparvata lugens*, *Cytorhinus Lividipennis* [18,19]. Volatiles from shallots and cucumbers invaded by *Thrips tabaci* can attract natural enemies such as *Orius similis* and *Amblyseius cucumeris* [20]; thus, it can be seen that releasing VOCs to attract natural enemies of herbivorous insects was an indirect defense strategy of plants.

Active components of VOCs can be used to develop insect attractants to trap insect pests, and to be made into repellents for controlling pests in the field, and can also be used to produce attractants for controlling natural enemy insects. Mixing plants that can release volatiles to trap or repel pests in the field can also achieve the effect of controlling pests and protecting crops [21,22,23]. Using plant volatiles, which are natural products, to control pests, has little impact on the ecological environment, and can achieve economic, social, and ecological benefits at the same time, which accords with the goal of sustainable pest control [24].

Herbivorous insects can perceive plants’ VOCs in the environment by their olfactory sense to distinguish host plants [25,26]. The process of insect olfactory sensation is inseparable from several key proteins, mainly including olfactory receptors (ORs), chemosensory proteins (CSPs), odorant-binding proteins (OBPs), odorant-degrading enzymes (ODEs), ionic receptors (IRs), and sensory neuron membrane proteins (SNMPs) [27]. The OBPs are the primary peripheral olfactory proteins that play critical roles in the odor detection of insects [28]. Elevated CO_2_ increased *F*. *occidentalis*’ selection for *P*. *vulgaris* via regulating the relative expression levels of *OBP1* and *OBP1-q* [29]. Transcriptome analysis showed that there were many differential OBP genes in *F. occidentalis* treated with different CO_2_ levels [30].

Studies have shown that the active VOCs for *F. occidentalis* included benzoic acid, salicylaldehyde, cinnamic acid, ethyl nicotinate, linalool, benzaldehyde, myrcene, o-anisaldehyde, myrcene, nerol, etc. [31,32] However, the research on *F. occidentalis* and VOCs mainly focuses on the evaluation of the effect of volatile monomers, and the research on component mixing is lacking. Some studies have shown that the effect of multi-component attractants in the field was often higher than that of a single-component attractant due to the influence of field background odor and other factors [33,34]. Therefore, based on screening monomer plant volatiles with a selective response to *F. occidental* by a four-arm olfactometer, the effects of binary and ternary mixtures of effective monomers were also further evaluated in this study for the purpose of making clear whether multi-component mixing can play a better role in pest control. At the same time, the effects of the above volatiles on the transcription level of OBPs were also measured, in order to preliminarily clarify the molecular mechanism of the effects of these VOCs from kidney bean plants on *F. occidentalis*. The ultimate goal is to develop some effective attractants or repellents based on the VOCs from plants for *F. occidentalis*, which could be used for the ecological control of *F. occidentalis*.

## 2. Results

### 2.1. Selection Behavior Response of F. occidentalis for Six Single VOCs of P. vulgaris

According to our previous identification of differential VOCs of *Phaseolus vulgaris* L. under different treatments [29], six volatile components of *P. vulgaris* were selected, and each volatile component was set to three concentration gradients: high, medium, and low. The behavioral response of *F. occidentalis* to the pure compounds corresponding to the six VOCs of *P. vulgaris* is shown in Figure 1. It can be seen that 1,3-dimethyl-4-ethyl benzene and 1,3-dimethyl benzene had no significant influence on the selection behavior of *F. occidentalis* under three different concentrations (*p* > 0.05, Figure 1). The high concentration of ethyl benzene, medium concentration of (E, E, E, E)-squalene, and low concentration of dioctyl phthalate had significant attraction effects for *F. occidentalis* (*p* < 0.05, Figure 1), and medium and high concentrations of 2,6-ditert-butyl-4-methyl phenol had significant repellent effects for *F. occidentalis* (*p* < 0.05, Figure 1). It was thus obvious that the behavioral selection of *F. occidentalis* depends on the type and concentration of the specific VOCs released by *P. vulgaris*.

### 2.2. Selection Behavior Response of F. occidentalis for Mixed VOCs of P. vulgaris

The single specific volatile compounds that had a significant attraction for *F. occidentalis* (including low concentration of dioctyl phthalate, medium concentration of (E, E, E, E)-squalene, and high concentration of ethyl benzene) were selected to make bi- and tri-mixtures, which are seen in Table 1. Treatment No. 1 had significant attraction effects on *F. occidentalis* (*p* < 0.05, Figure 2). In addition, treatments No. 2 and No. 3 also had significant attraction effects on *F. occidentalis* (*p* < 0.01, Figure 2), while treatment No. 4 had no significant effect on the selection behavior of *F. occidentalis* (*p* > 0.05, Figure 2).

### 2.3. Effect of VOCs of P. vulgaris on the Transcription Level of OBP Genes in F. occidentalis

According to previous research results from our laboratory, there were nine differential expression genes of OBPs including *LOC113205297*, *LOC113209726*, *LOC113212498*, *LOC113203140OBP2*, *LOC113205302*, *LOC113202849*, *LOC113205269*, *LOC113203279*, and *LOC113214975* [30]. The transcription levels of the above OBP genes that may be involved in the selection behavior of *F. occidentalis* were determined by qRT-PCR. For the OBP gene of *LOC113205297*, treatments E, B, and EDB had significant effects on the relative expression level. In detail, treatments E and B significantly down-regulated the transcription level of *LOC113205297* in *F. occidentalis* by 19.66% and 65.01% (*p* < 0.05), respectively, and the treatment EDB significantly up-regulated the relative expression level of *LOC113205297* by 42.97% (*p* < 0.001) compared to the control treatment (Figure 3A).

For the OBP gene of *LOC113209726*, treatments E, B, and D significantly affected its relative expression level (*p* < 0.01, Figure 3B), while treatments ED, EB, DB, and EDB did not significantly affect its transcription level. Specifically, treatments E and D significantly up-regulated the relative expression level of *LOC113209726* by 61.95% and 37.88% respectively (*p* < 0.001), while treatment B significantly down-regulated the relative expression level of *LOC113209726* by 22.66% compared to the control treatment (*p* < 0.01, Figure 3B).

For the OBP gene of *LOC113212498*, treatments D, B, and DB significantly down-regulated its relative expression level by 36.64% (*p* < 0.001), 33.64% (*p* < 0.001), and 21.76% (*p* < 0.01), respectively, and treatments EB and EDB significantly up-regulated its relative expression level by 17.63% (*p* < 0.05) and 28.24% (*p* < 0.001), respectively, compared to the control treatment (Figure 3C). And treatments E and D significantly down-regulated the relative expression level of *LOC113203140OBP2* by 27.67% and 30.30%, respectively (*p* < 0.001). The treatment EB significantly up-regulated the relative expression level of *LOC113203140OBP2* by 16.54% (*p* < 0.05) compared to the control treatment (Figure 3D).

The relative expression level of *LOC113214975* was up-regulated in all treatments (Figure 3E) and that of *LOC113205302* was down-regulated in all treatments (Figure 3F). Treatments E, D, B, EB, DB, and EDB significantly up-regulated the relative expression level of *LOC113214975* by 57.51%, 95.70%, 15.52%, 15.08%, 125.30%, and 172.45%, respectively (*p* < 0.05; Figure 3E). All the treatments E, D, B, ED, EB, DB, and EDB significantly down-regulated the relative expression level by 49.97%, 32.45%, 20.26%, 35.55%, 26.66%, 40.40%, and 36.83%, respectively (*p* < 0.05; Figure 3F).

The relative expression level of *LOC113202849* showed different trends under different treatments (Figure 3G). The treatment E significantly down-regulated the transcription level of *LOC113202849* by 23.02% (*p* < 0.001), and treatments D, ED, and EB significantly up-regulated the transcription level by 15.02%, 15.85%, and 70.68% compared to the control treatment (*p* < 0.05), respectively (Figure 3G). Moreover, all the treatments E, D, B, ED, EB, DB, and EDB had no significant effects on the transcription level of *LOC113205269* and *LOC113203279* (Figure 3H,I).

### 2.4. Correlation Analysis of the Selection Rate with the Transcript Expression Levels of OBP Genes in F. occidentalis Adults

The Pearson analysis showed that the selection rate of *F. occidentalis* adults was positively correlated with the relative expression levels of *LOC113209726* and *LOC113214975*, and negatively correlated with the relative expression levels of *LOC113205297*, *LOC113203140OBP2*, *LOC113205302*, and *LOC113202849* under treatment E (Figure 4A). And the selection rate was positively correlated with the relative expression levels of *LOC113205297*, *LOC113209726*, *LOC113214975*, and *LOC113202849*, and negatively correlated with the relative expression levels of *LOC113212498*, *LOC113203140OBP2*, and *LOC113205302* under treatment D (Figure 4B). Moreover, the selection rate was positively correlated with the relative expression level of *LOC113214975*, and negatively correlated with the relative expression levels of *LOC113205297*, *LOC113212498*, and *LOC113205302* under treatment B (Figure 4C).

Under treatment ED, the selection rate of *F. occidentalis* adults was positively correlated with the relative expression levels of *LOC113209726*, *LOC113203140OBP2*, and *LOC113214975*, and negatively correlated with the relative expression level of *LOC113205302* (Figure 4D). And the selection rate was positively correlated with the relative expression levels of *LOC113212498*, *LOC113203140OBP2*, *LOC113214975*, and *LOC113202849*, and negatively correlated with the relative expression level of *LOC113205302* under treatment ED (Figure 4E). Moreover, the selection rate was positively correlated with the relative expression levels of *LOC113209726*, *LOC113203140OBP2*, and *LOC113214975*, and negatively correlated with the relative expression levels of *LOC113212498* and *LOC113205302* under treatment DB (Figure 4F). Furthermore, the selection rate was positively correlated with the relative expression level of *LOC113212498*, and negatively correlated with the relative expression levels of *LOC113205269* and *LOC113203279* under treatment EDB (Figure 4G).

## 3. Discussion

The VOCs released by plants are important links to mediate chemical communication between species, including plants and insects, and plants and natural enemies of pests, which can regulate pollination, seed germination, and protect themselves from phytophagous insects, parasites, viruses, etc. [35,36] They can be potentially used in ecological pest prevention [37,38]. Many researchers have made great progress in this topic research on the biosynthesis, metabolism, function, and interaction with environmental and herbivorous insects in recent years [34,39]. Most of the volatile substances released by plants belong to alkanes, aromatic derivatives, alcohols, etc., all of which form odor mixtures through specific proportions, thus affecting the identification, selection, and orientation of herbivorous insects to their host plants [40]. For example, corn seedlings had obvious attraction effects on the sixth instar larvae of *Spodoptera frugiperda* [5]. Pepper volatiles can repel *Musca domestica* [41]. The composition and concentration of plant volatiles were different in different growth stages or tissues [42], and this different response of herbivorous insects to different concentrations of volatiles might be beneficial for insects to find suitable host plants accurately. Tang et al. found that ethyl acetate, butanone, α-pinene, and ethanol had attractive effects on *Kallima inachus* at a specific concentration [43]. Cis-jasmone had a significant attraction to *Helicoverpa armigera* at a specific concentration [44]. The olfactory responses of female *F. occidentalis* to ethyl nicotinate and *ρ*-anisaldehyde at concentrations of 0.05, 0.1, 0.5, and 1 μg/μL were significantly higher than those to liquid paraffin. The strongest response was toward 0.05 μg/μL ethyl nicotinate or *ρ*-anisaldehyde [45]. Thus, it can be seen that *F. occidentalis* may have different olfactory responses to different concentrations of *P. vulgaris* volatiles. According to above results, mainly combined with our previous identification of differential VOCs of *P. vulgaris* under different treatments [29], six kinds of VOCs (ethyl benzene, 1,3-dimethyl benzene, 1,3-dimethyl-4-ethyl benzene, (E, E, E, E)-squalene, 2,6-ditert-butyl-4-methyl phenol, and dioctyl phthalate) with three different concentration gradients were selected to explore the behavioral response of *F. occidentalis* to them. Our results showed that four volatiles had significant effects on the behavioral response of *F. occidentalis* at a specific concentration: 1.27 μL/μL of (E, E, E, E)-squalene, 0.32 μL/μL of dioctyl phthalate, and 8.22 μL/μL of ethyl benzene had obvious attraction effects on *F. occidentalis*, while 1.07 μL/μL and 2.14 μL/μL of 2,6-ditert-butyl-4-methyl phenol had obvious repellent effects on *F. occidentalis* compared to the control treatment. This indicated that the different VOCs produced by *P. vulgaris* under different treatments may affect the olfactory response of *F. occidentalis*. Although these VOCs that affect the selection behavior of *F. occidentalis* might be different from those in other plants at present [31,32,38,45], which may also expand the types of VOCs that affect the behavior selection of *F. occidentalis*.

Plant volatiles form odor mixtures in a specific proportion, which affects the process of phytophagous insects’ identification, selection, and orientation of their host plants, which is manifested as attracting or repelling pests [46,47]. For example, the mixtures with (E)-β-caryophyllene, (E)-4,8-dimethyl-1,3,7-nonatriene, and (E)-β-farnesene released from grapevine has a strong attraction to *Lobesia botrana*, but increasing or decreasing the components in them weakens or even removes the attraction effect [48]. For *F. occidentalis* females, the 1 μg/μL mixtures of ethyl nicotinate and *ρ*-anisaldehyde had a stronger attractive effect compared to liquid paraffin, but a high concentration caused a weak reaction [45]. In order to determine whether the mixed VOCs of *P. vulgaris* can significantly affect the selection behavior of *F. occidentalis*, we mixed the above-mentioned volatiles with an attraction effect at specific concentrations (12.7 μL/mL (E, E, E, E)-squalene, 3.2 μL/mL dioctyl phthalate, and 82.2 μL/mL ethyl benzene) in binary and ternary ways (treatments ED, EB, DB, and EDB). It was shown that treatments ED, EB, and DB have significant attraction effects on *F. occidentalis*, while the three mixed components EDB had no significant attraction effects on *F. occidentalis*. It was obvious that plant VOCs with an attractive effect after mixing might not improve the attractive effect, even having the opposite effect. Similar to where two kinds of volatile monomers with an attractive effect after mixing had an obvious repellent effect for *Megalurothrips usitatus* [49]. Although the mechanism of antagonism after the mixing of various single volatiles was not clear, the antagonistic effect of mixed monomers of plant VOCs has often been reported, such as by Diabate et al. who studied the behavioral response of *M. usitatus* to nonhost plant volatiles, and four kinds of volatile monomers, including three kinds of non-significant repellent effects and an attractant, produced significant repellent effect after blending [50]. Except for thrips, in the research of repelling and attracting green pea seeds by *Callosobruchus maculatus*, it was found that the quinary, quaternary, and ternary formulas of plant volatiles are not as attractive as the binary of plant volatiles for males [51]. It was reported that the attraction or repellent ability of a single compound is limited, while binary or ternary mixtures have achieved a better attraction or repellent effect in the field [52]. Therefore, in the effective mixing experiment of plant volatiles, the more the same-effect (attracting or repelling) plant volatiles were mixed, the better the effect was.

Odorant-binding proteins play critical roles in the insect olfactory system, and are responsible for capturing and transporting outside odorants through hydrophilic lymph to olfactory receptors [53,54]. The function of OBPs in VOCs’ recognition has been widely confirmed [55,56,57]. Nine differential OBP genes from transcriptome analysis results of *F. occidentalis* under different treatments [30] were selected to measure their relative expression levels for the purpose of exploring the reasons for the above phenomenon. In our study, the relative expression levels of *LOC113203279* and *LOC113205269* were not affected by all treatments, but the transcription level of *LOC113214975* was up-regulated universally, and *LOC113205302* was significantly down-regulated under all treatments. This indicated that the functions of *LOC113214975* and *LOC113205302* in olfactory recognition were conservative and critical, although they were different from what has previously been reported [29,57,58]. It was more likely that there may be antagonism between *LOC113214975* and *LOC113205302*. For other genes of OBPs, the relative expression levels showed different trends under different treatments. We suspected that the same OBP gene may also respond to mixtures and single odorants with different sensitivities, or the number of olfactory receptor neurons that bind to the mixtures may differ [59]. Moreover, correlation analysis showed that there may be synergy or antagonism between them. In general, the monomers and mixtures of the tested VOCs had significant effects on the relative expression levels of more than two OBP genes, which would lead to the up-regulation of some OBP genes and the down-regulation of other OBP genes, thus promoting or inhibiting the sensitivity of insects to the compounds corresponding to odor-binding proteins.

## 4. Materials and Methods

### 4.1. Insect Rearing

The colony of western flower thrips, *F. occidentalis*, was collected from Shandong Academy of Agricultural Sciences in Jinan, Shandong Province of China, and reared on the *P. vulgaris* leaves in insect-rearing cages for more than 10 generations. All the thrips used in this study were grown in artificial climate chambers with the following constant conditions: 27/25 ± 1 °C (day/night), 65 ± 5% humidity, and 14:10 h light/dark photoperiod.

### 4.2. Testing Compounds Preparation

The pure compounds corresponding to the volatile components of *P. vulgaris* leaves were purchased from Beijing Spectrum Analysis Technology in Beijing, China (Table 2). These pure compounds to be tested were diluted with N-hexane (Analytical Pure, Aladdin), and the configured concentration was determined according to the volume ratio of *P. vulgaris* leaves in our previous research [29].

### 4.3. Preparation of Uni-Lure and Multi-Lure

The preparation method of one-element lure used N-hexane as solvent, and the volatiles to be tested were respectively prepared into reagents with high, medium, and low concentrations (Table 2). Filter paper was cut into a 1 cm square as a sustained-release carrier, and 100 μL of the prepared reagent with the specified concentration and the N-hexane were dripped onto the filter paper to make a lure core and a blank control. The core was made and used before the experiment.

The preparation method of multiple lures was basically the same as that of one-element lures. According to the test results of *F. occidentalis* on single volatile matter from *P. vulgaris*, the mixtures were made with attractive single volatiles of 3.2 μL/mL dioctyl phthalate, 12.7 μL/mL (E, E, E, E)-squalene, and 82.2 μL/mL ethyl benzene for *F. occidentalis*. The formulas are shown in Table 1.

### 4.4. Selection Assays of F. occidentalis Adults for the Special VOCs

The effects of special VOCs on the selection behavior of *F. occidentalis* were quantified by using a four-chamber olfactometer (PSM4-150; Nanjing Pusen Instrument Co., Ltd., Nanjing, China). The diagonal ends of the four-chamber olfactometer were set as the “treatment areas” for the control and lure treatments. An 8 W fluorescent lamp was placed above the four-arm motherboard and the flow meter was adjusted to deliver a consistent airflow of 200 mL/min to all two sides. Thirty *F. occidentalis* adults within 3 days of new emergence were selected randomly and starved for 4 h, and then released to the center of the four-arm motherboard to observe their selection behavior. If the sampled *F. occidentalis* adults reached the “nesting area” of one arm within 20 min, the treatment corresponding to that arm was considered as the choice of the released *F. occidentalis* adults. Those *F. occidentalis* adults that did not reach any nesting area within 20 min after release were considered non-responders (i.e., no choice). Three replicates per experiment were set up. The four-arm olfactometers were rotated horizontally by 90° every time the experiment is repeated to avoid the position influence. In order to avoid biases in the behavioral observations between tests, the air compressor was turned off for 10 min and wiped with anhydrous alcohol after each test. The intake pipe was also exchanged after each test. And all tests were carried out in a clean, uniform, well-ventilated, and relatively closed laboratory. The *F. occidentalis* adults tested for host selection were collected for the following gene expression analysis of OBP genes.

### 4.5. RNA Extraction, cDNA Synthesis, and qRT-PCR Analysis

Thirty *F. occidentalis* adults were collected from each biological replicate of each treatment (including three biological replicates) for RNA isolation to analyze the gene transcript expression levels of OBP genes. Total RNA was isolated from the whole body of *F. occidentalis* adults by using the TRIzol^®^ reagent (Invitrogen, Carlsbad, CA, USA). The concentration and quality of samples were determined by using the NanoDropTM spectrophotometer (Thermo Scientific, Waltham, MA, USA) and 1.5% agarose gel electrophoresis. The cDNA synthesis was carried out with 100 ng of total RNA by using the PrimeScript^TM^ RT reagent Kit with gDNA Eraser (Takara, Osaka, Japan). Reverse transcriptase reactions were performed in a reaction volume of 20 μL. The qRT-PCR was performed with a 7500 real-time PCR detection system (Applied Biosystems, Foster City, CA, USA) by using 1× SYBR^®^ Premix Ex Taq^TM^ (TaKaRa, Osaka, Japan), 2 μL cDNA products (diluted from 20 μL to 200 μL with RNase-free water) and 0.2 μM primers in a final reaction volume of 20 μL. The specific primers for the OBP genes according to transcriptome results of *F. occidentalis* and the reference genes *β-actin* and *rpl32* [30] are listed in Table 3. The genes’ expression levels were quantified following the 2^−ΔΔCt^ normalization method, respectively [60]. The relative expression level was represented as the fold changes by comparing the samples of different special VOC treatments. Three technical replicates were also performed on each sample of cDNA.

### 4.6. Statistical Analysis

The statistical analysis of the data in this study was performed by the SPSS 20.0 software (IBM Corporation, Armonk, NY, USA). All measured index values are represented as mean values and their standard errors (SE). Independent-sample *t* test was used to analyze the effects of six pure compounds corresponding to volatile components of *P. vulgaris* plants on the selection behavior of *F. occidentalis*. One-way ANOVA was used to analyze the effect of different lures treatments on the transcript expression of OBP genes in *F. occidentalis*.

## 5. Conclusions

In conclusion, we found that three kinds of VOCs released by *P. vulgaris* have an attractive effect on *F. occidentalis* and their binary mixtures can also attract *F. occidentalis*. However, their ternary mixture had no effect on *F. occidentalis*. There may be synergistic or antagonistic effects between OBPs, which affect the olfactory response of *F. occidentalis* to VOCs. And the synergistic or antagonistic effects between OBPs will need to be proven in the future. We should fully consider the possible interaction (synergy or antagonism) between these plant VOCs. Meanwhile, the ability of insects’ odor-binding proteins to bind plant VOCs should also be noted to determine the transport capacity of VOCs in olfactory receptors, which can decrease or avoid antagonism between components in the design of mixed formulas, and also design the components with synergistic effects. Finally, the effective concentrations and the best compound components were determined through systematic experimental research to avoid antagonism as much as possible so as to obtain the best attraction or repellent effect.

## Figures and Tables

**Figure 1 ijms-24-12789-f001:**
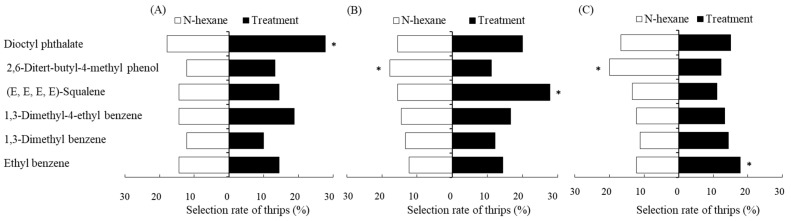
Effects of artificial standard samples (ASSs) corresponding to six specific volatile organic compounds (VOCs) from *Phaseolus vulgaris* L. plants with low (**A**), medium (**B**), and high (**C**) concentrations on the selective behavior of *Frankliniella occidentalis.* (Note: *, represented significant difference by the Independent-sample *t* test at *p* < 0.05.)

**Figure 2 ijms-24-12789-f002:**
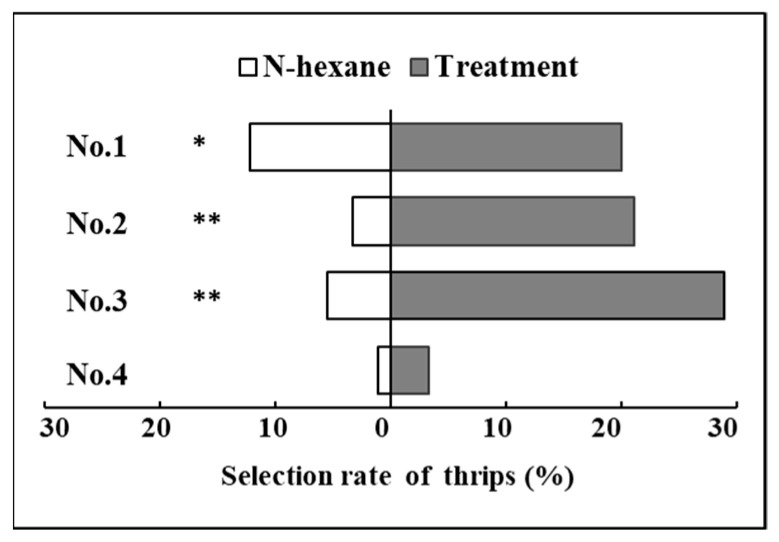
The selectivity rate (%) of *F*. *occidentalis* to mixtures with three ASSs corresponding to six specific VOCs from *P*. *vulgaris* that have significant attraction effects on the behavioral response of *F*. *occidentalis.* (Note: * and ** represented significant difference by the Independent-sample *t* test at *p* < 0.05 and *p* < 0.01, respectively. The mixture treatments of No. 1, No. 2, No. 3, and No. 4 were reported in Table 1).

**Figure 3 ijms-24-12789-f003:**
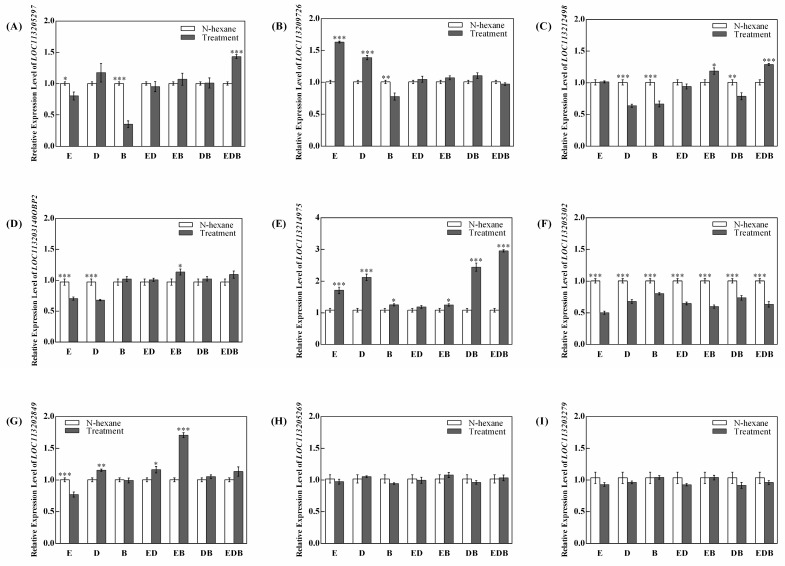
Relative expression levels of the odorant-binding protein genes (OBPs) of *LOC113205297* (**A**), *LOC113209726* (**B**), *LOC113212498* (**C**), *LOC113203140OBP2* (**D**), *LOC113214975* (**E**), *LOC113205302* (**F**), *LOC113202849* (**G**), *LOC113205269* (**H**), and *LOC113203279* (**I**) in *F*. *occidentalis* under control of N-hexane and plant volatiles treatments. (Note: E = 12.7 μL/mL (E, E, E, E)-squalene; D = 3.2 μL/mL dioctyl phthalate; B = 82.2 μL/mL ethyl benzene; ED = mixing of E and D; EB = mixing of E and B; DB = mixing of D and B; EDB = mixing of E, D, and B; *, **, *** represented significant difference by the independent-sample *t* test at *p* < 0.05, *p* < 0.01, *p* < 0.001, respectively.)

**Figure 4 ijms-24-12789-f004:**
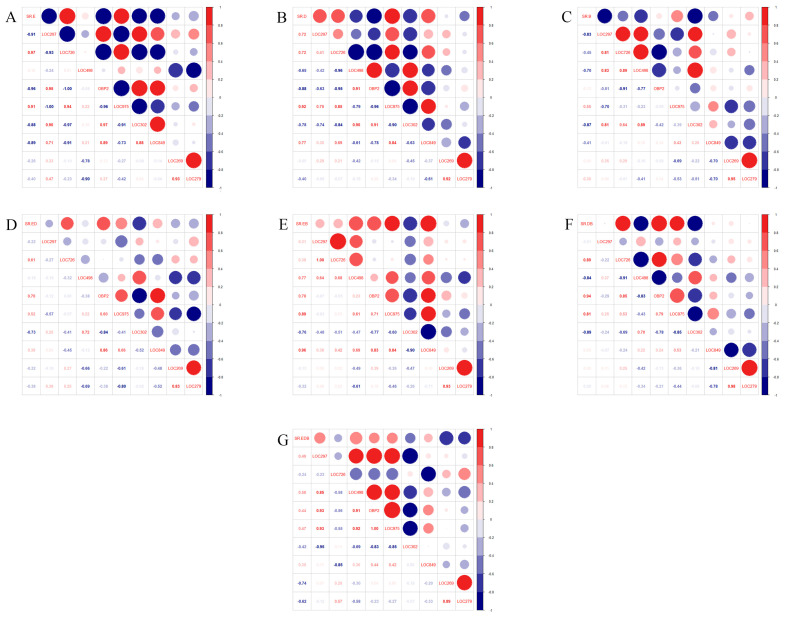
Analysis of Pearson’s correlation coefficient between the selection rate (SR) and transcription levels of OBP genes (LOC297, LOC726, LOC498, OBP2, LOC975, LOC302, LOC849, LOC269, and LOC279) under treatments with E (**A**), D (**B**), B (**C**), ED (**D**), EB (**E**), DB (**F**), and EDB (**G**). (Note: SR: selection rate of *F*. *occidentalis* adults; LOC297: *LOC113205297*; LOC726: *LOC113209726*; LOC498: *LOC113212498*; OBP2: *LOC113203140OBP2*; LOC975: *LOC113214975*; LOC302: *LOC113205302*; LOC849: *LOC113202849*; LOC269: *LOC113205269*; LOC279: *LOC113203279*. The scale color of the filled squares indicates the strength of the correlation (r) and whether it is negative (blue) or positive (red). The correlation is stronger when the number corresponding to the color and size of circles is closer to 1 or −1.)

**Table 1 ijms-24-12789-t001:** Mixtures with three artificial standard samples (ASSs) in corresponding to six specific VOCs from *Phaseolus vulgaris* L. that have significant effects on the selection behavior response of *Frankliniella occidentalis*.

Treatments	12.7 μL/mL (E, E, E, E)-Squalene	3.2 μL/mL Dioctyl Phthalate	82.2 μL/mL Ethyl Benzene
No. 1	+	+	−
No. 2	+	−	+
No. 3	−	+	+
No. 4	+	+	+

Note: “+” stands for adding the compound. “−” represents that the compound is not added.

**Table 2 ijms-24-12789-t002:** The relative concentrations (%) of specific volatile organic compounds (VOCs) from the leaves of *P. vulgaris* used to test the behavior response of *F. occidentalis*.

Volatile Types	VOCs	Structure Formula	CASNumber	Concentration (%)
Low	Medium	High
Aromatic hydrocarbons	Ethyl benzene	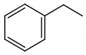	100-41-4	5.28	6.75	8.22
1,3-Dimethyl benzene	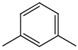	108-38-3	36.57	40.71	44.85
1,3-Dimethyl-4-ethyl benzene	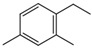	874-41-9	1.32	1.89	2.46
Olefins	(E, E, E, E)-Squalene	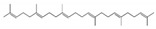	111-02-4	0.26	1.27	2.27
Phenols	2,6-Ditert-butyl-4-methyl phenol	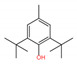	128-37-0	0	1.07	2.14
Esters	Dioctyl phthalate	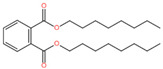	117-81-7	0.32	0.52	0.72

**Table 3 ijms-24-12789-t003:** qRT-PCR primers for odorant-binding protein (OBP) genes of *F. occidentalis*.

Primers		Sequence	Description
*LOC113205297*	Forward	AGCTAACCCCCTCACTGCTA	Odorant-binding protein genes
	Reverse	AGGATGCACTTGACGTAGCC	
*LOC113209726*	Forward	CATGGCTGAGTACCAGACGG	
	Reverse	GCCTGTAGCTCCTCCCAAAG	
*LOC113212498*	Forward	CACTCCCATCCCGACAAAGG	
	Reverse	AACCTCTGTCGCCACGATTT	
*LOC113203140OBP2*	Forward	AGTCTGATTCCGAGCTCCCT	
	Reverse	GTCTCTGTCTGCAGCGAAGT	
*LOC113214975*	Forward	GGTTGACGCCGATATGTTGC	
	Reverse	CTGCATCCCCTTAGACGACC	
*LOC113205302*	Forward	GCTACTGTCGCCAGTGAACA	
	Reverse	TTGGAAGCCTCTCTCTTTCGC	
*LOC113202849*	Forward	GCGTCGAAATAGAGCCCAGT	
	Reverse	CACCTGTCGCTTATGCCTGA	
*LOC113205269*	Forward	ATGAACCACGATGCCTGTGT	
	Reverse	CGTCATCCGTCATAGTGCCA	
*LOC113203279*	Forward	ATCAAAGAGTGCGCAGACCA	
	Reverse	CTTCCGAATGCTTCAGCACG	
*β-actin*	Forward	ACGACTTACAACTCCATCA	Housekeeping genes
	Reverse	AGTGCCTCCAGACAAAA	
*rpl32*	Forward	CTGGCGTAAACCTAAGGGTATTGA	
	Reverse	AAGCACCTTCTTGAACCCAGTC	

## Data Availability

The data obtained in this study have been presented “as is” in at least one of the figures or tables embedded in the manuscript.

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
