# Peer review of "Selection Behavior and OBP-Transcription Response of Western Flower Thrips, *Frankliniella occidentalis*, to Six Plant VOCs from Kidney Beans"

_ijms, 2023, doi:10.3390/ijms241612789_

Round 1

Reviewer 1 Report

The study is interesting, but there are some shortcomings that need to be corrected before the manuscript will be accepted for publication.

The novelty of the study is missing in the Introduction section

In the Results and Discussions section, there is no justification for the criteria that were the basis for the selection of VOC compounds.

It is not clearly highlighted why only three VOC compounds were chosen for the preparation of single and multi-lures, respectively.

Author Response

Comments to the Author

The study is interesting, but there are some shortcomings that need to be corrected before the manuscript will be accepted for publication.

  1. Reviewer's comment: The novelty of the study is missing in the Introduction section.

Response: According to your request, we have revised the content related to novelty in the Introduction section of the MS (line 116-125). The novelty of the MS was to evaluate the attraction of binary and ternary mixtures to Frankliniella occidentalis and the role of OBPs genes in this process.

  1. Reviewer's comment: In the Results and Discussions section, there is no justification for the criteria that were the basis for the selection of VOC compounds.

Response: Thank you for your constructive suggestions, we supplemented the justification for the criteria that were the basis for the selection of VOC compounds in the Results and Discussions section in line 133-135 (in the Results) and 374-389 (especially as described in more detail in the Discussions).

  1. Reviewer's comment: It is not clearly highlighted why only three VOC compounds were chosen for the preparation of single and multi-lures, respectively.

Response: Thanks for your suggestion. We re-revised the reasons for only three VOC compounds were chosen for the preparation of single and multi-lures in line 402-414.

Reviewer 2 Report

The manuscript “Selection behavior and OBPs-transcription response of western flower thrips, Frankliniella occidentalis to six plant VOCs from kidney beans” reports the laboratory investigation of the response in selective behavior of western flower thrips driven by single compound or mixture of common bean VOC. Moreover, to clarify the molecular mechanism of the effects of these VOCs the authors determined the transcription level of OBPs.

The topic of the article is interesting, and the laboratory experiments are well conducted. The introduction reports the state of the art on the topic discussed, although a bit lacking on the part about plant-mediated tri-trophic interactions and biological pest control. The introduction section is redundant in some parts to rewrite more fluently. In the manuscript, authors can find all suggestions and comments useful to revise this section.

In the "results" section I suggest some changes in the abbreviation of the name of treatments and review the significance levels comparing what is reported in the figures and the text.

The materials and methods are exhaustive, but I suggest reviewing the inclusion of the tables in the text, especially as regards the results section.

I suggest changing the section "Conclusions" to "Discussion” and improving the comparison of the results obtained in this manuscript with those reported in other articles, to demonstrate the importance of the results obtained over those already published. Comments/suggestions are included in the manuscript.

The conclusion is missing. The authors should reiterate the objective of their study reported in the manuscript, the importance of their results, tell the reader what contribution the study has made to the existing literature, and point out the problems and questions remaining.

Revise the supplementary materials folder, including only what is not listed in the manuscript, and add captions.

In conclusion, I suggest you review the manuscript taking into account all comments reported in the attached pdf file.

Author Response

Comments from the reviewer #2

Comments to the Author

The manuscript “Selection behavior and OBPs-transcription response of western flower thrips, Frankliniella occidentalis to six plant VOCs from kidney beans” reports the laboratory investigation of the response in selective behavior of western flower thrips driven by single compound or mixture of common bean VOC. Moreover, to clarify the molecular mechanism of the effects of these VOCs the authors determined the transcription level of OBPs.

  1. Reviewer's comment: The topic of the article is interesting, and the laboratory experiments are well conducted. The introduction reports the state of the art on the topic discussed, although a bit lacking on the part about plant-mediated tri-trophic interactions and biological pest control. The introduction section is redundant in some parts to rewrite more fluently. In the manuscript, authors can find all suggestions and comments useful to revise this section.

Response: We are extremely grateful to you for your detailed revision of our MS, which has benefited us a lot. According to your suggestions in Introduction section, we have revised, simplified and added content in Introduction again. You can find this content in line 52, 54-62, 68, 75, 77-78, 80, 106-109, 112 and 117-125. Of course, according to your suggestions, we have also revised other sections of the MS. Because there were too many modifications, we did not list the modified lines one by one.

  1. Reviewer's comment: In the "results" section I suggest some changes in the abbreviation of the name of treatments and review the significance levels comparing what is reported in the figures and the text.

Response: Thank you for your meaningful suggestion. We have revised the name of treatments (in line 157-166), and also checked and corrected the significant differences (line 163). Meanwhile according to your suggestions in the Results, we revised the errors and redundancies in the Results section without modified lines.

3. Reviewer's comment: The materials and methods are exhaustive, but I suggest reviewing the inclusion of the tables in the text, especially as regards the results section.

Response: Thanks for your valuable proposal. We have modified the Names and Notes of Figures and Tables (in line 147-149, 152, 169-170, 172-178, 231-233, 262, 286-288, 301-304 and 351-352), and moved Figure 2 in the Materials and Methods section to Results section 2.2 as Figure 1 (in line175-178 and 301-304), and the order of other Tables had also been revised in line 286.

  1. Reviewer's comment: I suggest changing the section "Conclusions" to "Discussion” and improving the comparison of the results obtained in this manuscript with those reported in other articles, to demonstrate the importance of the results obtained over those already published. Comments/suggestions are included in the manuscript.

Response: According to your suggestions, we changed the "Conclusions" to "Discussion” and re-added the "Conclusions" section. At the same time, we have made extensive revisions to the Discussion section according to your revision opinions. You can find it in line 353-528.

5. Reviewer's comment: The conclusion is missing. The authors should reiterate the objective of their study reported in the manuscript, the importance of their results, tell the reader what contribution the study has made to the existing literature, and point out the problems and questions remaining.

Response: Thanks for your valuable proposal. We re-added the Conclusions section in line 516-528.

6 Reviewer's comment: Revise the supplementary materials folder, including only what is not listed in the manuscript, and add captions.

Response: Thanks for your suggestion. There were no supplementary Figures and Tables. The original data of this manuscript has been uploaded to the ijms system. So this section is “Not applicable”. You can find it in line 529-530.

7. Reviewer's comment: In conclusion, I suggest you review the manuscript taking into account all comments reported in the attached pdf file.

Response: We greatly appreciate again your valuable advices on our manuscript. We have made a lot of revisions to the MS. And most of the modified lines have been mentioned above. Other modification was tiny and miscellaneous, so we did not list them.

Round 2

Reviewer 1 Report

Agree with the revised manuscript.

Please replace at line 19-23, 387-388:

Ethyl benzene with ethyl benzene

1,3-Dimethyl benzene with 1,3 dimethyl benzene

1,3-Dimethyl-4-ethylbenzene with dimethyl-4-ethylbenzene

(E, E, E, E)-Squalene with  (E, E, E, E)-squalene

2,6-Ditert-butyl-4-methyl phenol with   2,6-ditert-butyl-4-methyl phenol

Dioctyl phthalate with dioctyl phtalate

The same changes must be done at line 392-394, 407, 410, 415

Author Response

Comments from the reviewer #1

Agree with the revised manuscript.

  1. Reviewer's comment: Please replace at line 19-23, 387-388: Ethyl benzene with ethyl benzene; 1,3-Dimethyl benzene with 1,3 dimethyl benzene; 1,3-Dimethyl-4-ethylbenzene with dimethyl-4-ethylbenzene; (E, E, E, E)-Squalene with (E, E, E, E)-squalene; 2,6-Ditert-butyl-4-methyl phenol with 2,6-ditert-butyl-4-methyl phenol; Dioctyl phthalate with dioctyl phthalate; The same changes must be done at line 392-394, 407, 410, 415.

Response: We are extremely grateful to you for your detailed revision of our MS, we have revised the names of all the above VOCs (in line in 19-23, 387-388, 392-394, 407, 410, 415), and corrected the names of VOCs in other parts of the MS. You can find them in line 138-143, 156-157, 232-233 and 290-291.
